# Completion Time Minimization for UAV-UGV-Enabled Data Collection

**DOI:** 10.3390/s22155839

**Published:** 2022-08-04

**Authors:** Zhijian Li, Wendong Zhao, Cuntao Liu

**Affiliations:** School of Communication Engineering, Army Engineering University, Nanjing 210007, China

**Keywords:** unmanned aerial vehicle–unmanned ground vehicle, path planning, data collection

## Abstract

In unmanned aerial vehicle (UAV)-enabled data collection systems, situations where sensor nodes (SNs) cannot upload their data successfully to the UAV may exist, due to factors such as SNs’ insufficient energy and the UAV’s minimum flight altitude. In this paper, an unmanned ground vehicle (UGV)-UAV-enabled data collection system is studied, where data collection missions are conducted by a UAV and a UGV cooperatively. Two cooperative strategies are proposed, i.e., collaboration without information interaction, and collaboration with information interaction. In the first strategy, the UGV collects data from remote SNs (i.e., the SNs that cannot upload data to the UAV) as well as some normal SNs (i.e., the SNs that can upload data to the UAV), while the UAV only collects data from some normal SNs. Then, they carry the data back to the data center (DC) without interacting with each other. In the second strategy, the UGV only collects data from remote SNs, while transmitting the collected data to the UAV at a data interaction point, then the data are carried back to the DC by the UAV. There are mobile data collection nodes on the ground and in the air, and the task is to find trajectories to minimize the data collection time in the data center. A collaborative strategy selection algorithm, combining a multi-stage-based SN association and UAV-UGV path optimization algorithm, is proposed to solve the problem effectively, where techniques including convex optimization and genetic algorithm are adopted. The simulation result shows that the proposed scheme reduces the mission completion time by 36% compared with the benchmark scheme.

## 1. Introduction

Unmanned aerial vehicles (UAVs) and unmanned ground vehicles (UGVs), as mobile actuators, are being deployed in many application areas, including disaster relief [1,2,3,4,5,6], surveillance in war zones [7], and energy-efficient data collection over wireless sensor networks (WSNs) [8]. Energy consumption is one important design aspect for WSNs since battery replacement or recharging for sensor nodes (SNs) is typically inconvenient, costly, and sometimes impossible. By employing a UAV or UGV as a mobile data collector, each SN can directly send its sensed data to it.

However, new challenges arise when a UAV or UGV performs data collection missions alone. To ensure the safety of UAVs and compliance with UAV traffic rules, UAVs are usually specified with a minimum safe flight altitude and no-fly zones [9]. SNs that cannot successfully upload their data to the UAV may exist, due to factors such as SNs’ insufficient energy and the UAV’s minimum flight altitude. We define these SNs as remote SNs. In addition, SNs may generate data on a regular basis [10], which makes it important to minimize the mission completion time for ongoing data collection missions so that subsequent missions can begin early. Although the UGV can collect remote SNs’ data nearby, it will take more time to complete the mission due to the slow movement speed. The UAV and UGV can complement each other, for example, the UAV can move quickly from one place to another and have line-of-sight (LoS) communication links with SNs. On the other hand, the UGV can carry larger payloads, closer to the target, and provide the ability to intervene in the environment, deploy SNs and communication equipment, etc. [11]. Therefore, we propose a UAV-UGV-enabled data collection system.

The path planning problem becomes a critical problem for the UAV-UGV-enabled data collection system. In addition, idle listening is one important cause of energy waste for many sensor network applications. To address this issue, the sleep/wake-up scheme has been introduced [12], which is able to adjust the ratio between the sleeping and awake time of each SN to minimize idle listening. In this article, we consider a sensor to be associated with the device it is awakened by. If each SN is only woken up and transmits data when UAV-UGV is closer, then the energy consumed by the SNs is reduced. However, at this point, UAV-UGV would need to move further to get closer to the SNs, which would result in a longer completion time. There is a fundamental trade-off between the completion time and the energy consumption of SNs.

Accordingly, this paper focuses on the path planning problem for the UAV-UGV-enabled data collection system. Our work makes the following three contributions:We propose a UAV-UGV-enabled data collection system with two cooperative strategies. We aim to minimize the mission completion time by jointly optimizing the UAV-UGV paths, as well as the SN association while ensuring that SNs can successfully upload their data with the remaining energy.To tackle the formulated problem, we propose a simple scheme in that UAV-UGV only collects data while staying near or above SNs. The problem is reduced to finding the optimal stop-over above or near the SNs and the time duration at each staying location, as well as the serving order among these locations.We propose a collaborative strategy selection algorithm, combining a multi-stage-based SN association and UAV-UGV path optimization algorithm, to solve the problem effectively, where techniques including convex optimization and genetic algorithm are adopted.

The rest of the paper is organized as follows. The next section presents a literature review. In Section 3, we introduce the system model and present the problem formulation. In Section 4, we present the proposed algorithm. In Section 5, we present simulation results. Finally, conclusions and future research directions are outlined.

## 2. Literature Review

UAV-UGV collaboration systems are increasingly used in both military and civilian applications because they allow coordinated actions to leverage the strengths of each to achieve a common goal [13]. The strong complementarity between UAVs and UGVs offers many new application prospects for UAV-UGV collaboration systems. Our focus is on cooperative path planning in sensor data collection. Path planning is the basis for the research of UAV-UGV-enabled data collection [14].

Considering the heterogeneous and complementary nature of UAVs and UGVs, two types of UAV-UGV-enabled data collection approaches have been extensively studied in the existing literature. The first one is to play separate functional roles. For example, the UGV is set up as a charging station for UAVs to improve the flying range. In this case, the UAV needs to periodically visit the stationary power station acted by the UGV and replenish energy via a battery replacement system, or the UGV acts as a mobile charging station and the UAV needs to periodically rendezvous with the UGV for resupply [15].

In [16], the authors investigate the use of UGVs as mobile charging stations to solve the energy-constrained problem of UAVs. To avoid the introduction of UGVs as charging stations leading to long mission cycles, the combination of classical techniques and modern evolutionary approaches is used to minimize the distance traveled by UGVs while satisfying the UAV charging requirements and full node coverage. In [17], the path planning problem of multiple UGVs as mobile charging stations for a UAV is investigated for a large-area distribution of sensors. An FCURP-MRS model is proposed, which ensures that UGVs can charge UAVs by generating multiple UGV and UAV paths to shorten the mission cycle and minimize the travel distance of UAVs. The authors of [18] investigated a scenario where a UGV acts as a mobile charging station for multiple small UAVs by jointly optimizing the trajectories of the UGV and the UAV to minimize the maximum flight time of multiple UAVs, while ensuring that charging can be completed for multiple UAVs. The authors of [19] considered the charging of a single energy-limited UAV by a UGV that can only travel on a defined grid road, with the UAV responsible for collecting data from sensors. A two-stage algorithm was used to solve this problem. Firstly, a set of charging station locations was found using the UAV range restrictions. Secondly, a mixed-integer linear programming problem was formulated to obtain the paths.

The second approach is to play the same functional role. There are significant differences in terms of detection accuracy, detection angle, and mechanical structure. By effectively coordinating UGVs and UAVs with the same functional role, data collection, inspection, and target tracking can be achieved. When performing data collection, reconnaissance, and tracking missions, a large area can be covered quickly with the UAV, and the UGV can get closer to the target and intervene in it.

In the data collection mission, the authors of [20] proposed a data collection scheme that uses UGVs as mobile cluster heads in WSNs to assist the UAV to save data collection time and energy for SNs. The ground mobile cluster head is responsible for aggregating sensor data together and the UAV obtains the required information from it. Finally, the route of the ground mobile cluster head is optimized according to the network density. In [21], a mixed-integer linear programming (MILP) algorithm was developed by considering the communication constraints between the UAV and UGV to ensure that the data collected by the UAV can be transmitted to the UGV instantaneously, which uses a branch-and-cut strategy to plan the paths of the UAV and the UGV. In [22], a UAV-UGV cooperative environmental monitoring system is proposed, in which both the UAV and UGV are used as mobile sensors. The UAV is responsible for monitoring the airborne atmospheric quality and the UGV is responsible for monitoring the ground-based atmospheric quality, and the system has been extensively tested in field experiments. A heterogeneous system of UAVs and UGVs with the mission of monitoring the terrain on a given path is presented in [23]. Both the UAV and UGV are equipped with cameras that monitor the terrain within their field of view, making terrain mapping more efficient and mapping data more accurate. Path planning for UAV-UGV collaboration systems can be well-generalized to the path length-based traveling salesman problem (TSP) [24] and can be solved by some classical heuristic algorithms.

In summary, most researchers assume that UAVs can always successfully receive data from SNs, which is not true. To ensure the safety of UAVs and compliance with UAV traffic rules, UAVs are usually specified with a minimum safe flight altitude and no-fly zones [9]. SNs that cannot successfully upload their data to the UAV may exist, due to factors such as SNs’ insufficient energy and the UAV’s minimum flight altitude. We define these SNs as remote SNs. In most studies, there is no joint mission assignment for the UAV and the UGV. In this paper, a UAV-UGV-enabled data collection system is studied, where data collection missions are cooperatively conducted by a UAV and a UGV to minimize the mission completion time by jointly optimizing the selection of the cooperative strategy, the SN association, and the UAV-UGV path.

## 3. Problem Description

As shown in Figure 1, one UAV and one UGV are assigned to collect data from K SNs. Some SNs cannot upload data due to their limited transmission power and the limited safe flight altitude of the UAV. Two cooperative strategies are proposed, i.e., collaboration without information interaction, and collaboration with information interaction. In the first strategy, the UGV collects data from remote SNs (i.e., the SNs that cannot upload data to the UAV) as well as some normal SNs (i.e., the SNs that can upload data to the UAV), while the UAV only collects data from some normal SNs. Then, they carry the data back to the data center (DC) without interacting with each other. In the second strategy, the UGV only collects data from remote SNs, while transmitting the collected data to the UAV at a data interaction point, then the data are carried back to the DC by the UAV. The goal is to minimize the mission completion time.

### 3.1. Problem Assumptions

(1)The position of SNs, the remaining energy, and the amount of data to be uploaded are known before the mission is performed.(2)To ensure the safety of the UAV, the UAV flies at a fixed safe altitude.(3)The UGV and UAV depart from the DC with sufficient energy to complete the mission.(4)The UAV and UGV collect the data in an orthogonal frequency channel, and SNs only wake up when they are scheduled for data transmission to save energy.

### 3.2. Problem Models

The notations that appear in the model are provided in Table 1.

The positions of the UAV and the UGV at the time slot n are denoted by $q_{UAV}[n]={\left(x_{UAV}[n],y_{UAV}[n],H\right)}^{T}$ and $q_{UGV}[n]={\left(x_{UGV}[n],y_{UGV}[n],0\right)}^{T}$, respectively. Then, the path of the UAV and the UGV can be denoted as $\left\{q_{UGV}[n]\right\}$ and $\left\{q_{UGV}[n]\right\}$. In addition, the deployment position of SNs is known in advance, and the coordinates of SN $s_{k}$ are expressed as $w_{k}={\left(x_{k},y_{k},0\right)}^{T}$. Assume that the channel between SN $s_{k}$ and UAV is a LoS channel. The channel gain from SN $s_{k}$ to UAV is:(1)$$h_{k,UAV}[n]=\frac{\rho_{0}}{{\left\|q_{UAV}-w_{k}\right\|}^{2}},\forall n,$$
where $\rho_{0}$ denotes the channel power gain at a reference distance of 1 m, then the SN to UAV upload rate (bits/s/Hz) can be obtained as:(2)$$R_{k,UAV}[n]=\log_{2}\left(1+\frac{P_{k}\rho_{0}}{\sigma^{2}{\left({\left\|q_{UAV}[n]-w_{k}\right\|}^{2}\right)}^{\frac{\alpha_{1}}{2}}}\right),\forall n,\forall k,$$
where $\sigma^{2}$ is additive Gaussian white noise (AWGN) power received by the receiver at the UAV and α is the path loss index. Similarly, the upload rate from SN $s_{k}$ to the UGV and the transmission rate from the UGV to the UAV are:(3)$$R_{k,UGV}[n]=\log_{2}\left(1+\frac{P_{k}\rho_{0}}{\sigma^{2}{\left({\left\|q_{UGV}[n]-w_{k}\right\|}^{2}\right)}^{\frac{\alpha_{2}}{2}}}\right),\forall n,\forall k$$
(4)$$R_{UGV,UAV}[n]=\log_{2}\left(1+\frac{P_{UGV}\rho_{0}}{\sigma^{2}{\left({\left\|q_{UGV}[n]-q_{UAV}[n]\right\|}^{2}\right)}^{\frac{\alpha_{1}}{2}}}\right),\forall n,\forall k$$

The minimum energy required by the SN $s_{k}$ to transmit its data to the UAV, i.e., the energy consumed by the SN to upload data when the UAV is hovering above the SN is: $E_{k}^{\min}=D_{k}P_{k}/R^{hover}$, where $R^{hover}=\log_{2}\left(1+P_{k}\rho_{0}/\sigma^{2}H^{2}\right)$. If $E_{k}<E_{\min}$, SN $s_{k}$ data can only be collected by the UGV.

Idle listening is one important cause of energy waste for many sensor network applications. To address this issue, the sleep/wake-up scheme has been introduced [12], which is able to adjust the ratio between the sleeping and awake time of each SN to minimize idle listening. Define $a_{m,k}\in \{0,1\},\forall u_{m}\in M,s_{k}\in K$. If SN $s_{k}$ is awakened by $u_{m}$ and transmits data to $u_{m}$, $a_{m,k}=1$, and inversely, $a_{m,k}=0$. Therefore, we can obtain $G_{m}=\{s_{k}|a_{m,k}=1,\forall k\},\forall m$. Using $\Gamma_{m}=\{\varepsilon_{1},\ldots ,\varepsilon_{\left|G_{m}\right|}\}$ to denote the label of the access order of $u_{m}$, the access order can be determined as $\left(s_{\varepsilon_{1}},\ldots ,s_{\varepsilon_{\left|G_{m}\right|}}\right)$. For example, if $G_{m}=\{s_{2},s_{4},s_{6}\}$, $\Gamma_{m}=\{4,2,6\}$, then the $u_{m}$’s access order is $\left(s_{4},s_{2},s_{6}\right)$. To ensure that the data can be uploaded with the remaining energy, the region of feasible staying locations for $u_{m}$ can be obtained as:(5)$$q_{m,k}[t]\in \left\{\frac{D_{k}P_{k}}{R_{k,m}}\leq E_{k},s_{k}\in G_{m},0\leq t\leq t_{k}\right\}$$

### 3.3. Problem Formulation

The SN association and UAV-UGV path are optimized to minimize the mission completion time. The mission completion time is denoted by $T_{K}$. When strategy A is used, TK=max Tm, conversely, $T_{K}=T_{UAV}$. The problem can be formulated as:(6)$$\underset{\left\{a_{k,m}\right\},\{q_{m}[n]\},\left\{\Gamma_{m}\right\}}{\min}T_{K},$$
(7)$$s.t.\;\;\;\left\|q_{m}[n+1]-q_{m}[n]\right\|/\delta_{t}\leq V_{\max}^{m},\forall m$$
(8)$$a_{m,k}\in \{0,1\},\forall k,m$$
(9)$$\sum_{m=1}^{M}a_{k,m}=1,\forall k$$
(10)$$q_{m,k}[t]\in \left\{\frac{D_{k}P_{k}}{R_{k,m}}\leq E_{k},s_{k}\in G_{m},0\leq t\leq t_{k}\right\},\forall k,m$$
(11)$$\sum_{t=0}^{t_{k}}R_{k,m}^{}(t)\geq D_{k},\forall k,m$$

The objective function (6) indicates that our goal is to minimize the time for data to reach the DC. Constraint (9) indicates that any SNs can only be associated with either the UAV or UGV. Constraints (10) and (11) indicate that the energy of the SNs is guaranteed to be sufficient to upload the data. This problem is NP-hard due to the presence of integer variables and the discrete nature of $\Gamma_{m}$.

## 4. Proposed Solution

We propose a simple scheme that UAV-UGV only collects data while staying near or above SNs, termed as staying mode (S-mode). For this mode, to find the optimized stop-over above or near the SNs and the access order among all locations, we propose an efficient algorithm by leveraging the min–max multiple traveling salesman problem (min–max m-TSP) and convex optimization techniques.

Define $q_{\varepsilon_{0}}=q_{\varepsilon_{|G_{m}|+1}}=q_{I}$. The mission completion time consists of the moving time and the collecting time. Since S-mode is used, the moving time and the collecting time are only related to the staying locations, and to minimize the moving time, both the UGV and the UAV move between staying locations at maximum speed. The moving time and the staying time can be obtained as $T_{m}^{f}=\sum_{l=1}^{|G_{m}|+1}\left\|q_{\varepsilon_{l}-}q_{\varepsilon_{l-1}}\right\|/V_{\max}^{m}$,$T_{m}^{h}=\sum_{l=1}^{\left|G_{m}\right|}\tau_{l}^{h}$, where $\tau_{l}^{h}=D_{l}/R_{m,l}$ represents the time used by $u_{m}$ to collect data from SN $s_{l},1\leq l\leq \left|G_{m}\right|$.

### 4.1. Two-Stage Optimization Algorithm for Strategy A

The problem is decoupled into two subproblems. In the first subproblem, the SN association $\left\{G_{m}\right\}$ and the access order $\left\{\Gamma_{m}\right\}$ are obtained by solving Equation (12). In the second subproblem, given the association $\left\{G_{m}\right\}$ and the access order $\left\{\Gamma_{m}\right\}$ obtained from the first subproblem, the staying locations, $q_{\varepsilon_{l}}^{}$, are optimized to minimize the mission completion time.

#### 4.1.1. Solving for the Association and the Access Order

In the first subproblem, given that $\left\{q_{\varepsilon_{l}}^{0}\right\}=\left\{w_{\varepsilon_{l}}^{}\right\}$, then the problem can be formulated as follows:(12)$$\underset{\left\{a_{m,k}\right\},\left\{G_{m}\right\},\left\{\Gamma_{m}\right\}}{\min}T_{K}$$
(13)s.t. (9),(10)
(14)$$T_{m}^{f}+T_{m}^{h}\leq T_{K},\forall m$$
(15)$$K_{UGV}\in G_{UGV}\in K$$

The association and access order can be obtained by solving the min–max m-TSP, and although the problem is NP-hard, various algorithms exist to efficiently find high-quality approximate solutions (e.g., genetic algorithms) [25].

#### 4.1.2. Optimizing the UAV-UGV Path

Since $\tau_{l}^{h}$ is a non-convex function, and the sum of a finite number of non-convex functions is still a non-convex function, therefore, Equation (6) is a non-convex optimization problem. To solve the problem, relaxation variables $\left\{z_{l}\right\}$ are introduced.
(16)$$B\log_{2}\left(1+\frac{\gamma_{\varepsilon_{l}}}{∥q_{\varepsilon_{l}}-w_{\varepsilon_{l}}{∥}^{2}}\right)\geq z_{\varepsilon_{l}},\forall s_{\varepsilon_{l}}\in G_{m},\forall m$$

Since the left-hand side of Constraint (16) is a convex function, Constraint (16) is a non-convex constraint. The left-hand side of the equal sign of Constraint (16) is a convex function, which can be subjected to a first-order Taylor expansion and then obtained as a concave lower bound [26], which can be obtained by giving the local point $q_{\varepsilon_{l}}^{r}$.
(17)$$R_{\varepsilon_{l}}\geq A_{\varepsilon_{l}}^{r}-I_{\varepsilon_{l}}^{r}\left({\left\|q_{\varepsilon_{l}}-w_{\varepsilon_{l}}\right\|}^{2}-{\left\|q_{\varepsilon_{l}}^{r}-w_{\varepsilon_{l}}\right\|}^{2}\right)≜R_{\varepsilon_{l}}^{lb}$$
(18)$$A_{\varepsilon_{l}}^{r}=B\log_{2}\left(1+\frac{P_{\varepsilon_{l}}\rho_{0}}{\sigma^{2}{\left(H^{2}+{\left\|q_{\varepsilon_{l}}^{r}-w_{\varepsilon_{l}}\right\|}^{2}\right)}^{\alpha_{1}/2}}\right)$$
(19)$$I_{\varepsilon_{l}}^{r}=\frac{(\alpha_{1}/2)P_{\varepsilon_{l}}\rho_{0}\log_{2}e}{\sigma^{2}\left({\left\|q_{\varepsilon_{l}}^{r}-w_{\varepsilon_{l}}\right\|}^{2}\right)\left({\left({\left\|q_{\varepsilon_{l}}^{r}-w_{\varepsilon_{l}}\right\|}^{2}\right)}^{\alpha_{1}/2}+\frac{P_{\varepsilon_{l}}\rho_{0}}{\sigma^{2}}\right)}$$
where $R_{\varepsilon_{l}}^{lb}$ is a linear function that is both convex and concave. The equation holds at $q_{\varepsilon_{l}}^{}=q_{\varepsilon_{l}}^{r}$, and $R_{\varepsilon_{l}}^{lb}$ has the same gradient as $R_{\varepsilon_{l}}^{}$ [25]. For any given local point $\left\{q_{\varepsilon_{l}}^{r}\right\}$ and the lower bound expression in Equation (17), the problem is approximated by:(20)$$\underset{\left\{a_{m,k}\right\},\left\{G_{m}\right\},\left\{\Gamma_{m}\right\},\{z_{l}\}}{\min}T_{K}$$
(21)$$s.t.\;T_{m}^{f}+T_{m}^{h}\leq T_{K},\forall m$$
(22)$$\frac{P_{\varepsilon_{l}}D_{\varepsilon_{l}}}{z_{\varepsilon_{l}}}\leq E_{\varepsilon_{l}},\forall s_{\varepsilon_{l}}\in G_{m},\forall m$$
(23)$$\tau_{\varepsilon_{l}}^{h}\geq \frac{D_{\varepsilon_{l}}}{z_{\varepsilon_{l}}},\forall s_{\varepsilon_{l}}\in G_{m},\forall m$$
(24)$$R_{\varepsilon_{l}}^{lb}\geq z_{\varepsilon_{l}},\forall s_{\varepsilon_{l}}\in G_{m},\forall m$$

Since the objective function and constraint are convex constraints, Equation (20) is a convex problem, which can be efficiently solved using standard convex optimization tools such as CVX [27]. The original non-convex problem can be solved by applying the SCA technique [28] to iteratively optimize Equation (20), updating the local points {qεlr} at each iteration until the objective function converges. The overall algorithm for strategy A is summarized in Algorithm 1.
**Algorithm 1.** Overall algorithm for strategy A1. Initialize {qεl0}={wεl};2. Solving (12) with a genetic algorithm, then obtain {Gm} and {Γm};3. r←0,{qεlr}←{qεl0};4. **while** (Obj(r)-Obj(r−1)≤tol)5.    Given {qεlr}, solve (20) and denote the optimal solution as {qεlr+1};6.    {qεlr}←{qεlr+1}, r←r+1;7. **end while**

### 4.2. Three-Stage Optimization Algorithm for Strategy B

The association has been provided in strategy B, $G_{\mathrm{UGV}}=K_{\mathrm{UGV}}$$G_{\mathrm{UAV}}=K-G_{\mathrm{UAV}}$. Then, the problem is decoupled into three subproblems. The first subproblem is to obtain the access order $\left\{\Gamma_{m}\right\}$. The second subproblem is to obtain the intersection location $\left\{q_{c}^{m}\right\}$. The access order and the intersection location are then found by iterative means. Then, the third subproblem is to optimize $\left\{q_{\varepsilon_{l}}\right\}$ and $\left\{q_{c}^{m}\right\}$. In strategy B, the mission completion time is:(25)$$T_{K}=T_{UAV}=\max\{T_{c}^{UAV},T_{c}^{UGV}\}+\tau_{UGV}^{h}+\left\|q_{c}^{UAV}-q_{I}\right\|/V_{\max}^{UAV}$$
(26)$$T_{c}^{m}=\frac{\left\|q_{c}^{m}-q_{\varepsilon_{m,|G_{m}|}}\right\|+\sum_{l=1}^{\left|G_{m}\right|}\left\|q_{\varepsilon_{m,l}}-q_{\varepsilon_{m,l-1}}\right\|}{V_{\max}^{m}}+\sum_{l=1}^{\left|G_{m}\right|}\tau_{l}^{h},\forall m$$
(27)$$\tau_{UGV}^{h}=\sum_{l=1}^{\left|G_{UGV}\right|}D_{l}/R_{UGV,UAV}$$

#### 4.2.1. Solving for the Association and the Access Order

The first subproblem is to obtain the access order by solving the TSP given $\varepsilon_{\left|G_{m}\right|}$. It can be solved directly by a variety of algorithms, including genetic algorithms.

#### 4.2.2. Solving for Intersection Location

In the second subproblem, the intersection location of the UAV and the UGV is obtained by substituting the results found in the first subproblem. Since the left-hand side of Constraint (27) is a convex function, as in solving the problem (20), the relaxation variables,x, are introduced and a first-order Taylor expansion is performed on $R_{UGV,UAV}$ to obtain $R_{UGV,UAV}^{lb}$ by giving the local point ${\left(q_{c}^{m}\right)}^{r}$. For any given local point $\left\{{\left(q_{c}^{m}\right)}^{r}\right\}$ and concave function $R_{UGV,UAV}^{lb}$, the problem is approximated as:(28)$$\underset{\left\{q_{c}^{m}\right\},x}{\min}T_{UAV}$$
(29)$$s.t.\;\{T_{c}^{UAV},T_{c}^{UGV}\}+\tau_{UGV}^{h}+\left\|q_{c}^{UAV}-q_{I}\right\|/V_{\max}^{UAV}\leq T_{UAV}$$
(30)$$\tau_{UGV}^{h}\geq \sum_{l=1}^{\left|G_{UAV}\right|}D_{l}/x$$
(31)$$R_{UGV,UAV}^{lb}\geq x$$

Since the objective function and constraint are convex constraints, Equation (28) is a convex problem, which can be solved efficiently using standard convex optimization tools.

#### 4.2.3. Optimizing the UAV-UGV Path and Intersection Location

Finally, the optimal staying location $\left\{q_{\varepsilon_{l}}\right\}$ and the interaction point $\left\{q_{c}^{m}\right\}$ are obtained by solving the third subproblem. The relaxation variables x and $c_{k}$ are introduced and a first-order Taylor expansion is performed on $R_{UGV,UAV}$ and $R_{\varepsilon_{l}}^{}$ at the given local points $\left\{{\left(q_{c}^{m}\right)}^{r}\right\}$ and $\left\{q_{k}^{r}\right\}$. We can obtain the following problem:(32)$$\underset{\left\{q_{c}^{m}\right\},\left\{q_{k}^{r}\right\},\left\{c_{k}\right\},x}{\min}T_{UAV}$$
(33)s.t. (29),(30),(31)
(34)$$P_{\varepsilon_{l}}\tau_{\varepsilon_{l}}^{h}\leq E_{\varepsilon_{l}},\forall s_{\varepsilon_{l}}\in G_{m},\forall m$$
(35)$$\tau_{\varepsilon_{l}}^{h}\geq \frac{D_{\varepsilon_{l}}}{c_{\varepsilon_{l}}},\forall s_{\varepsilon_{l}}\in G_{m},\forall m$$
(36)$$R_{\varepsilon_{l}}^{lb}\geq c_{\varepsilon_{l}},\forall s_{\varepsilon_{l}}\in G_{m},\forall m$$

The objective function and constraints are convex, and the problem is convex. Equation (32) can be solved directly using convex optimization tools. The algorithm for strategy B is obtained and is shown in Algorithm 2.
**Algorithm 2.** Overall algorithm for strategy B1. kUGV=1**,**
T=∞**;**2. **while (**kUGV≤|GUGV|**)**3.    kUAV=14.    **while (**kUAV≤|GUAV|**)**5.      ε|GUGV|←GUGV(kUGV)**;**6.      ε|GUAV|←GUAV(kUAV)**;**7.      Solve TSP with a genetic algorithm to obtain the order of service {Γm};8.      Substitute {Γm},{qcm}0←wεGugv| and obtain the interaction point {qcm} by solving the convex problem (28)9.      **if**
TUAV<T10.        T←TUAV,{qcm}0←{qcm},{Γm0}←{Γm};11.       **end if**12.     kUAV←kUAV+1**;**13.     **end while**14.     kUGV←kUGV+1;15.    **end while**16.    Substitute {qcm}0 and {Γm0} into (32), and obtain {qεl} and {qcm}.

### 4.3. The Collaborative Strategy Selection Algorithm

Based on the results obtained in the previous two subsections, the overall algorithm for solving Equation (6) is proposed. By solving Equation (12) to obtain TUGV and TUAV, if TUGV≤TUGV+(τUGVh)min, the UAV need not receive data from the UGV, and strategy A is adopted, where (τUGVh)min is the time taken by the UGV to transmit the data to the UAV hovering above the UGV; conversely, if TUGV>TUAV+(τUGVh)min and Gugv,k=Kugv, then strategy B is adopted. The overall algorithm for Equation (6) is shown in Algorithm 3.
**Algorithm 3.** Overall algorithm1. Initialize {qεm,l0}={wεm,l};2.   Solving (12) with a genetic algorithm, obtain TUGV and TUAV;3.   **if**
TUGV≤TUAV+(τUGVh)min4.     Execution of Algorithm 1.5.   **else**6.     Execution of Algorithm 2.7.   **end if**

## 5. Simulation and Results

In this section, the performance of the proposed algorithm is evaluated. K = 15 SNs were uniformly and randomly distributed in a square area with a side length equal to 0.8 km. It is assumed that all SNs have the same transmit power. Assume that the remaining energy of the remote nodes is $2/3E_{\min}$ and the sufficient energy of the remaining SNs is $2E_{\min}$. For comparison, two strategies are simulated in this paper. The main parameters are shown in Table 2.

To demonstrate the effectiveness of our proposed scheme, we compared it to three benchmarks, namely the minimum staying time benchmark, the fixed interaction benchmark, and the non-synergistic benchmark.

In the minimum staying time benchmark, the UAV/UGV moves to the staying locations of its associated SNs with the maximum speed by solving the min–max m-TSP and stays there for a sufficient duration to complete the data collection.In the fixed interaction benchmark, the interaction point area is set to be midway between the center of the remote SNs and the center of the normal SNs.In the uncooperative benchmark, the UGV is only responsible for collecting data from the remote SNs and bringing the data back itself.

Figure 2 shows the UAV-UGV paths and associations for missions with different data sizes. It can be observed that as the $D_{k}$ increases, the UAV needs to spend more time on data collection as the SN requires more communication resources, when the advantage of the UGV arriving close to the data collection is more obvious, as expected. For example, compare Figure 2a,c. In Figure 2a, the SN has fewer data to upload, and the time difference between the UAV and the UGV for data collection is small. The mission completion time depends more on the moving time. In Figure 2c, the SN has more data to upload, the time difference between the UAV and the UGV for data collection is larger, and the mission completion time depends more on the data collection time. Therefore, when the amount of data is small, the UAV is responsible for more nodes. Otherwise, make the UGV responsible for more nodes. Similarly, compare Figure 2b,d. In Figure 2b, it is important to try to avoid the disadvantage of the UGV in terms of movement speed, and therefore to make the distance traveled by the UGV as small as possible, allowing the UAV to move further to receive the data from the UGV and then bring the data back. In Figure 2d, the UGV will collect the data in less time, compensating for the lack of movement speed, while the UAV takes time to collect the data and therefore does not need to UAV to receive the data from the UGV, and instead has the UGV wait on the return path of the UAV.

Figure 3 depicts the mission completion time versus the amount of data uploaded by SNs. It can be observed that as the amount of uploaded data increases, the mission completion time increases. When $D_{k}\leq 9$ Mbits, the performance of the proposed solution is close to that of strategy B, while when $D_{k}>9$ Mbits, the performance of the proposed scheme is close to that of strategy A. This is to be expected. Combine this with the trajectory analysis in Figure 2. When $D_{k}\leq 9$ Mbits, the mission completion time is more dependent on the moving time. Comparing Figure 2a,b, to reduce the disadvantage of the slow UGV movement speed, it is important to make the distance traveled by the UGV as small as possible, so when executing strategy A, only the UGV is allowed to collect data from the remote SNs. When executing strategy B, the UGV is made to move as small a distance as possible. When $D_{k}>9$ Mbits, the amount of data to be uploaded by the SN is high, and the mission completion time is more dependent on the data collection time. Comparing Figure 2c,d, to reduce the UAV’s disadvantage in data collection, the UGV has time to collect data from normal SNs after the remote SNs are collected. In strategy B, the UGV has more time to move to a position that is conducive to reducing the distance traveled by the UAV after collecting data from remote SNs. Since it is more advantageous for the UGV to be nearby for data collection, the use of strategy A to allow the UGV to collect data from more SNs yields better results. Compared with the uncooperative benchmark scheme, the proposed scheme reduces the task completion time by 36% on average.

Figure 4 depicts the mission completion time versus the number of insufficient-energy SNs. Strategy A is preferred when the number of remote nodes is low when the UGV must access fewer SNs and the UGV can collect from more SNs. When the number of remote SNs is higher, strategy B is preferred, because the UGV must access more SNs, and the UAV can quickly complete its mission. Therefore, try to keep the UGV execution time as low as possible and let the UAV receive data from the UGV. As the number increases again, the amount of data collected by the UGV increases, and the time to transmit the data to the UAV exceeds that for the UGV to bring back, so strategy A is chosen as superior. Moreover, when the number of remote SNs is low, the mission completion time of strategy B tends to decrease as the number of remote SNs increases. This is because the UGV will finish collecting the data it is responsible for in a short time and then move to the interaction point to wait for the UAV. After all, the UAV is responsible for more SNs. The waiting time of the UGV is enough to collect data from normal SNs. As the number of remote SNs increases and the number of SNs responsible for the UAV decreases, the time for the UAV to reach the intersection will advance, while the UGV can still reach the intersection before the UAV. As the number of remote SNs increases, the mission completion time will have a decreasing process. Compared with the uncooperative benchmark scheme, the proposed scheme reduces the task completion time by 23% on average.

Figure 5 depicts the relationship between UAV energy consumption and the amount of data uploaded by SNs. When $D_{k}\leq 9$ Mbits, the energy consumption of the UAV in strategy B is smaller than that in strategy A. This is because the UAV has an advantage over the UGV in saving more time when the amount of data to be collected is less, so the UAV will collect data from more SNs. When $D_{k}>9$ Mbits, the energy consumption of the UAV in strategy B is greater than that in strategy A. This is because, when the amount of data to be collected is larger, the UGV takes less time, so it will let the UGV move further to collect data from more SNs. The solution proposed in this paper allows the UGV/UAV to take full advantage of its advantages by choosing the strategy to speed up the mission completion time at the cost of consuming more energy of the UAV or the UGV.

Figure 6 depicts the relationship between the degree of aggregation of the remote nodes and the mission completion time. Where the degree of aggregation of the remote nodes is represented by the average distance from the remote nodes to the geometric center of all the remote nodes, the result is obtained by randomly taking 20% of the energy-deficient SNs for simulation several times and then taking the average value. When the remote nodes are all distributed in a small area, the mission completion time for strategy A is shorter. As the degree of aggregation decreases, the UGV takes more time to move between the remote nodes. When the time taken by the UGV to perform data collection is close to the time taken by the UAV, strategy B is chosen to allow the UAV to fly further to receive data from the UGV. Compared with the uncooperative benchmark scheme, the proposed scheme reduces the task completion time by 33% on average.

## 6. Conclusions

In this paper, the data collection mission in UAV-UGV-supported WSNs was investigated to minimize the completion time by jointly optimizing the path, association, and cooperative strategy selection, while ensuring that each SN can successfully upload data with the remaining energy. Two strategies were proposed, the corresponding solution algorithms were presented, and the strategy selection algorithm was proposed. Numerical results showed that the proposed scheme achieved a significant performance improvement over the benchmark scheme. In future work, the problem of minimizing the mission completion time in the more general cases, i.e., when the UAV and the UGV can collect and transmit data while moving, will be explored.

## Figures and Tables

**Figure 1 sensors-22-05839-f001:**
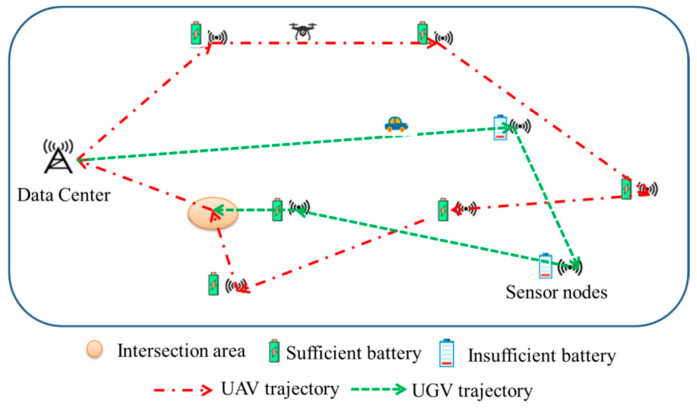
System model.

**Figure 2 sensors-22-05839-f002:**
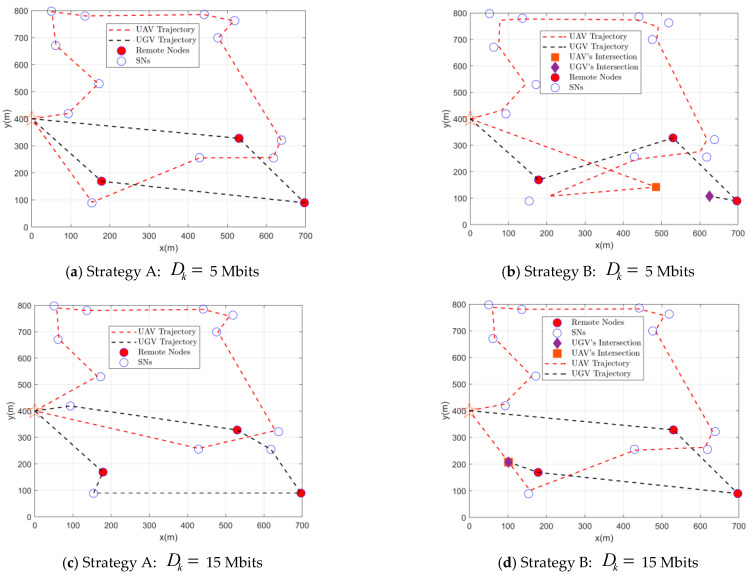
UAV and UGV trajectories.

**Figure 3 sensors-22-05839-f003:**
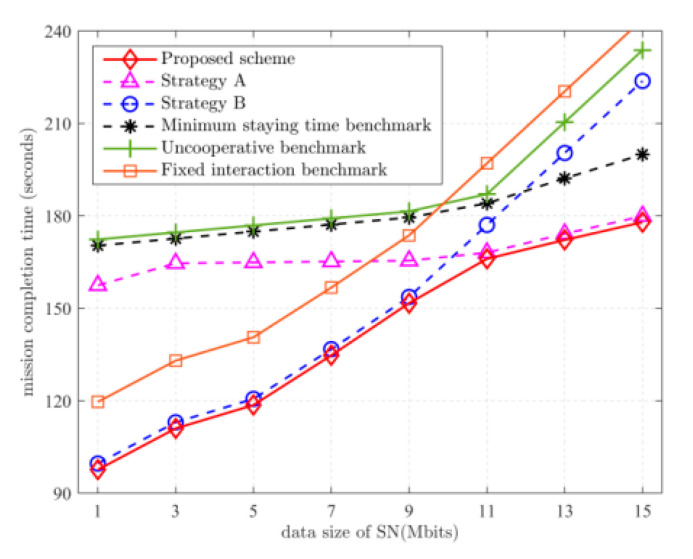
Relationship between mission completion time and $D_{k}$.

**Figure 4 sensors-22-05839-f004:**
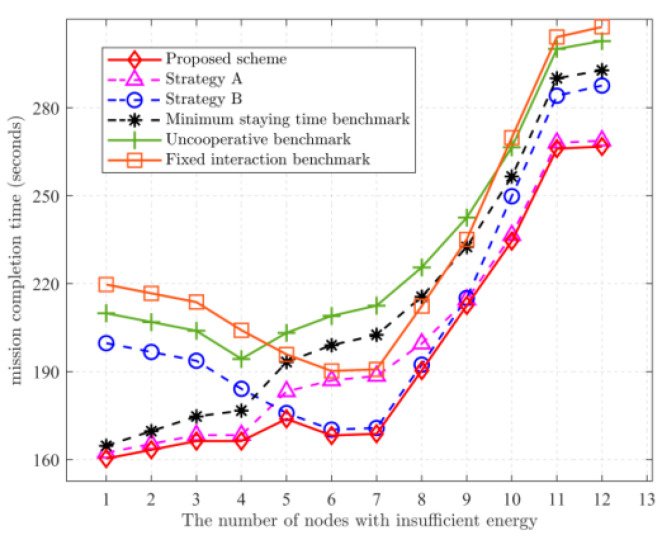
Relationship between mission completion time and the number of remote nodes. $D_{k}=12$ Mbits.

**Figure 5 sensors-22-05839-f005:**
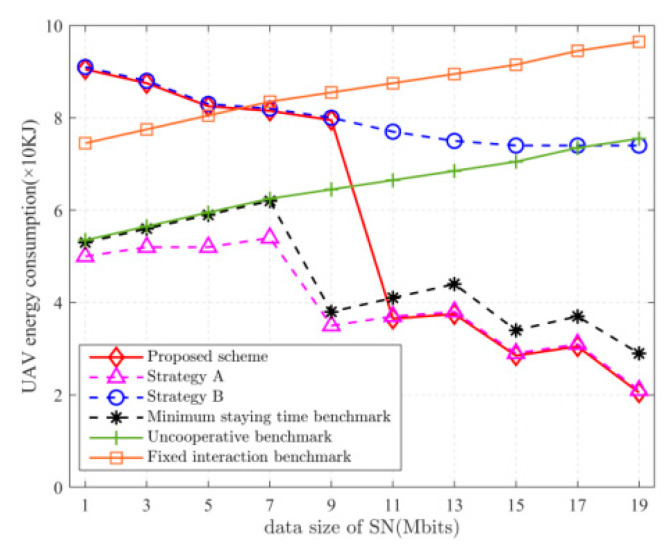
Relationship between UAV energy consumption and $D_{k}$.

**Figure 6 sensors-22-05839-f006:**
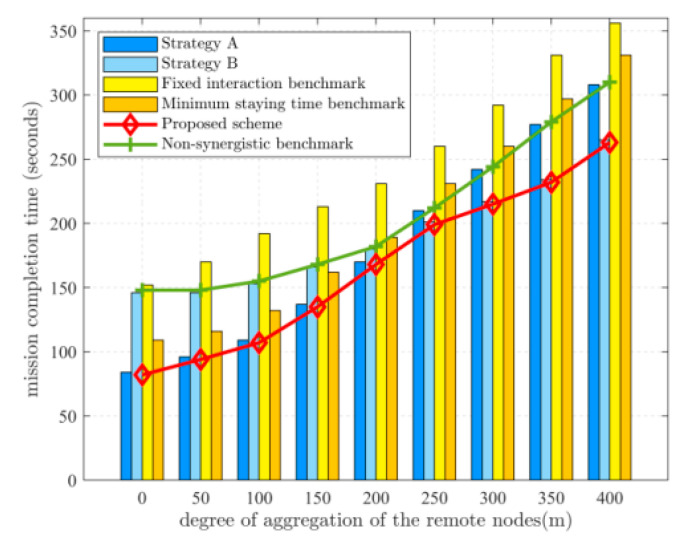
Relationship between mission completion time and the degree of aggregation of the remote nodes. $D_{k}=12$ Mbits.

**Table 1 sensors-22-05839-t001:** Model notations.

Notation Type		Notation Description
Sets	$K=\left\{s_{k},1\leq k\leq K\right\}$	The set of all sensors
	$M=\left\{u_{m},1\leq m\leq 2\right\}$	The set of UAV and UGV
	$G_{m}=\{s_{k}|a_{m,k}=1,\forall k\}$	The set of sensors associated with $u_{m}$
	$K_{UGV}=\{s_{k}|E_{k}<E_{k}^{\min},\forall k\}$	The set of remote SNs
Parameters	$q_{I}={\left(x_{I},y_{I}\right)}^{T}$	The horizontal position of the DC
	$q_{m}[n]={\left(x_{m}[n],y_{m}[n]\right)}^{T}$	The horizontal position of $u_{m}$
	$q_{c}^{m}={\left(x_{c}^{m},y_{c}^{m}\right)}^{T}$	The position of $u_{m}$ when interacting with data
	$V_{\max}^{m}$	The maximum movement speed of $u_{m}$
	$v_{m}[n]$	The movement speed of $u_{m}$ in time slot *n*
	H	The UAV flight height
	$w_{k}={\left(x_{k},y_{k},0\right)}^{T}$	The position of the SN $s_{k}$
	$D_{k}$	The amount of data to be uploaded by the SN $s_{k}$
	$E_{k}$	The remaining energy of the SN $s_{k}$
	$E_{k}^{\min}$	The energy required for the SN $s_{k}$ to upload data to the UAV hovering above it
	$P_{k}$	The transmitting power of the SN $s_{k}$
	$P_{\mathrm{UGV}}$	The transmitting power of the UGV
	$T_{m}$	The time for the UAV to return to the DC
	$T_{K}$	The time for all data to reach the DC
	$T_{m}^{f}$	The time taken by $u_{m}$ to move
	$T_{m}^{h}$	The time taken by $u_{m}$ to collect data
	$\Gamma_{m}$	The order of access to $u_{m}$
	$\left(\varepsilon_{1},\varepsilon_{2},\ldots ,\varepsilon_{\left|G_{m}\right|}\right)$	The label arrangement for $u_{m}$ to access SNs
Decision variables	$a_{m,k}$	Binary. If SN $s_{k}$ is associated with $u_{m}$, $a_{m,k}=1$.

**Table 2 sensors-22-05839-t002:** Simulation parameters.

Symbols	Numerical Values	Symbols	Numerical Values
H	100 m	B	1 MHz
$V_{\max}^{uav}$	50 m/s	$V_{\max}^{ugv}$	10 m/s
$\sigma^{2}$	−110 dBm	$\rho_{0}$	−60 dB
$P_{UGV}$	20 W	$P_{k}$	0.1 W

## Data Availability

The data presented in this study can be requested from the authors.

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
