# Peer review of "Completion Time Minimization for UAV-UGV-Enabled Data Collection"

_sensors, 2022, doi:10.3390/s22155839_

Round 1

Reviewer 1 Report

This paper proposes an Unmanned Ground Vehicle (UGV)-Unmanned Aerial Vehicle (UAV)-enabled cooperative data collection system. In this system, two cooperative strategies, including collaboration without information interaction and collaboration with information interaction, are developed and switched autonomously. The selection of cooperative strategy, sensor note association and UAV-UGV path are treated as a completion time minimisation problem and optimised. Simulation results indicate that the proposed scheme reduces the mission completion time by 36% compared with the benchmark scheme.

This paper is well-structured and easy to read. However, there are some grammar errors such as the “UAV-UGV collaboration system” on page 2 line 84 should be “UAV-UGV collaboration systems”. Please go over the whole article and current them. In addition, there are a few points that need to be clarified:

1. The definition of the wk should be mentioned earlier. 

2. The meaning of the SN association is not clear

3. Figure 2 and Figure 5 demonstrate the same conclusion. I suggest changing the vertical axis of Figure 5 as the system energy consumption. 

4. The conclusion “Simulation result shows that the proposed scheme reduces the mission completion time by 36% compared with the benchmark scheme.” Need to be clarified in the context of the “Simulation and Results” part.

Reviewer 2 Report

Dear Authors,

the article entitled: Completion Time Minimization for UAV-UGV Enabled Data Collection presents an Unmanned Aerial Vehicle (UAV) and an Unmanned Ground Vehicle (UGV) enabled data collection system, where data collection missions are conducted by an UAV and an UGV cooperatively. It is very innovative because the strong complementarity between UAVs and UGVs offers many new application prospects for UAV-UGV collaboration system such as, for example, a cooperative path planning in sensor data collection.

All chapters (abstract, introduction, literature review, problem description, proposed solution, simulation and results, as well as conclusions) are well described and they do not raise any doubts. In terms of the literature review is not complex (23 positions), all of which are papers from recognised scientific journals, such as: IEEE Communications Surveys and Tutorials, IEEE Internet of Things Journal, Proceedings of the IEEE, and others. Moreover, I would like to point out that the papers cited are related to the subject of this article (UAV-UGV, path planning and data collection). However, in the publication make the following change:

• I propose to extend the literature in the first sentence of the introduction, related to the applications of UAV and UGV platforms such as, for example:

1. DÄ…browski, P.S.; Specht, C.; Specht, M.; Burdziakowski, P.; Makar, A.; Lewicka, O. Integration of Multi-source Geospatial Data from GNSS Receivers, Terrestrial Laser Scanners, and Unmanned Aerial Vehicles. Can. J. Remote Sens. 2021, 47, 621–634.

2. Hua, C.; Niu, R.; Yu, B.; Zheng, X.; Bai, R.; Zhang, S. A Global Path Planning Method for Unmanned Ground Vehicles in Off-road Environments Based on Mobility Prediction. Machines 2022, 10, 375.

3. Mei, A.; Zampetti, E.; Di Mascio, P.; Fontinovo, G.; Papa, P.; D’Andrea, A. ROADS—Rover for Bituminous Pavement Distress Survey: An Unmanned Ground Vehicle (UGV) Prototype for Pavement Distress Evaluation. Sensors 2022, 22, 3414.

4. Specht, M.; Stateczny, A.; Specht, C.; Widźgowski, S.; Lewicka, O.; Wiśniewska, M. Concept of an Innovative Autonomous Unmanned System for Bathymetric Monitoring of Shallow Waterbodies (INNOBAT System). Energies 2021, 14, 5370.

To sum up, after taking into account the above amendments (minor revision), I suppose that this article is suitable for publication in the Sensors.

Reviewer 3 Report

The paper investigates useful and interesting problem of wireless data collection from a wireless sensor network using mobile nodes. However, the problem formulation and description appears a bit too convoluted. The paper could be revised to simplify how it is presented. Please consider the following suggestions.

1. Abstract could formulate the problem in simpler terms. Thus, there are mobile data collection nodes on the ground and in the air, and the task is to find trajectories to minimize the data collection time in the data center. 

2. The contributions listed on p. 2: they are indeed 3, the last one - simulations could be removed from the bullet point list, and left and as a paragraph. The sentence on lines 66-68 is unclear - is not that obvious that the trajectory is described by line segments and that to minimize the time, the speed should be maximum? In addition, 'staying' is not a proper term, please consider something like 'a stop-over above the node' or similar.

3. Line 136: better explain why data may not always be successfully collected from the sensor. This is a crucial part of the problem.

4. In Figure 1, it may be useful to add green bars to battery icon, if it is full. This is obviously a minor issue.

5. Problem assumptions: is it realistic that the remaining energy and amount of data to be uploaded are known beforehand? For the latter, perhaps if the amount of data is always constant. However, how does the sensor node knows when to wake up? If the wake up schedule is known, then it is a critical factor to consider in optimizing the paths.

6. Assuming channel capacity to model data collection is unnecessary complication. The problem of optimizing the paths for two type of mobile nodes is already complicated enough.

7. Taylor expansion: about which point? And if that point changes every time? 

8. More importantly, why there is no energy optimization considered? Even if sensors have low battery, their data still needs to be collected. 

9. Why the optimization problem is convex/non-convex? Note that convex optimization and genetic algorithm are two completely different things - problem type vs. algorithm type.

Round 2

Reviewer 3 Report

The authors have addressed all my comments.